# COMMUNICATION ALGORITHMS VIA DEEP LEARNING

**Hyeji Kim\*, Yihan Jiang†, Ranvir Rana\*, Sreeram Kannan†, Sewoong Oh\*, Pramod Viswanath\***
University of Illinois at Urbana Champaign\*, University of Washington†

## ABSTRACT

Coding theory is a central discipline underpinning wireline and wireless modems that are the workhorses of the information age. Progress in coding theory is largely driven by individual human ingenuity with sporadic breakthroughs over the past century. In this paper we study whether it is possible to automate the discovery of decoding algorithms via deep learning. We study a family of sequential codes parametrized by recurrent neural network (RNN) architectures. We show that creatively designed and trained RNN architectures can decode well known sequential codes such as the convolutional and turbo codes with close to optimal performance on the additive white Gaussian noise (AWGN) channel, which itself is achieved by breakthrough algorithms of our times (Viterbi and BCJR decoders, representing dynamic programing and forward-backward algorithms). We show strong generalizations, i.e., we train at a specific signal to noise ratio and block length but test at a wide range of these quantities, as well as robustness and adaptivity to deviations from the AWGN setting.

## 1 INTRODUCTION

Reliable digital communication, both wireline (ethernet, cable and DSL modems) and wireless (cellular, satellite, deep space), is a primary workhorse of the modern information age. A critical aspect of reliable communication involves the design of codes that allow transmissions to be robustly (and computationally efficiently) decoded under noisy conditions. This is the discipline of coding theory; over the past century and especially the past 70 years (since the birth of information theory (Shannon, 1948)) much progress has been made in the design of near optimal codes. Landmark codes include convolutional codes, turbo codes, low density parity check (LDPC) codes and, recently, polar codes. The impact on humanity is enormous – every cellular phone designed uses one of these codes, which feature in global cellular standards ranging from the 2nd generation to the 5th generation respectively, and are text book material (Richardson & Urbanke, 2008).

The canonical setting is one of point-to-point reliable communication over the additive white Gaussian noise (AWGN) channel and performance of a code in this setting is its gold standard. The AWGN channel fits much of wireline and wireless communications although the front end of the receiver may have to be specifically designed before being processed by the decoder (example: intersymbol equalization in cable modems, beamforming and sphere decoding in multiple antenna wireless systems); again this is text book material (Tse & Viswanath, 2005). There are two long term goals in coding theory: $(a)$ design of new, computationally efficient, codes that improve the state of the art (probability of correct reception) over the AWGN setting. Since the current codes already operate close to the information theoretic "Shannon limit", the emphasis is on *robustness* and *adaptability* to deviations from the AWGN settings (a list of channel models motivated by practical settings, (such as urban, pedestrian, vehicular) in the recent 5th generation cellular standard is available in Annex B of 3GPP TS 36.101.) (b) design of new codes for multi-terminal (i.e., beyond point-to-point) settings – examples include the feedback channel, the relay channel and the interference channel.

---
\*H. Kim, R. Rana and P. Viswanath are with Coordinated Science Lab and Department of Electrical Engineering at University of Illinois at Urbana Champaign. S. Oh is with Coordinated Science Lab and Department of Industrial and Enterprise Systems Engineering at University of Illinois at Urbana Champaign. Email: {hyejikim,rbrana2,swoh,pramodv}@illinois.edu (H. Kim, R. Rana, S.Oh, and P.Viswanath)

†Y. Jiang and S. Kannan are with Department of Electrical Engineering at University of Washington. Email: yihanrogerjiang@gmail.com (Y. Jiang), ksreeram@uw.edu (S. Kannan)

Progress over these long term goals has generally been driven by individual human ingenuity and, befittingly, is sporadic. For instance, the time duration between convolutional codes (2nd generation cellular standards) to polar codes (5th generation cellular standards) is over 4 decades. Deep learning is fast emerging as capable of learning sophisticated algorithms from observed data (input, action, output) alone and has been remarkably successful in a large variety of human endeavors (ranging from language (Mikolov et al., 2013) to vision (Russakovsky et al., 2015) to playing Go (Silver et al., 2016)). Motivated by these successes, we envision that deep learning methods can play a crucial role in solving both the aforementioned goals of coding theory.

While the learning framework is clear and there is virtually unlimited training data available, there are two main challenges: ($a$) The space of codes is very vast and the sizes astronomical; for instance a rate 1/2 code over 100 information bits involves designing $2^{100}$ codewords in a 200 dimensional space. Computationally efficient encoding and decoding procedures are a must, apart from high reliability over the AWGN channel. ($b$) Generalization is highly desirable across block lengths and data rate that each work very well over a wide range of channel signal to noise ratios (SNR). In other words, one is looking to design a family of codes (parametrized by data rate and number of information bits) and their performance is evaluated over a range of channel SNRs.

For example, it is shown that when a neural decoder is exposed to nearly 90% of the codewords of a rate 1/2 polar code over 8 information bits, its performance on the unseen codewords is poor (Gruber et al., 2017). In part due to these challenges, recent deep learning works on decoding known codes using data-driven neural decoders have been limited to short or moderate block lengths (Gruber et al., 2017; Cammerer et al., 2017; Dörner et al., 2017; O'Shea & Hoydis, 2017). Other deep learning works on coding theory focus on decoding known codes by training a neural decoder that is initialized with the existing decoding algorithm but is more general than the existing algorithm (Nachmani et al., 2016; Xu et al., 2017). The main challenge is to restrict oneself to a class of codes that neural networks can naturally encode and decode. In this paper, we restrict ourselves to a class of *sequential* encoding and decoding schemes, of which convolutional and turbo codes are part of. These sequential coding schemes naturally meld with the family of recurrent neural network (RNN) architectures, which have recently seen large success in a wide variety of time-series tasks. The ancillary advantage of sequential schemes is that arbitrarily long information bits can be encoded and also at a large variety of coding rates.

Working within sequential codes parametrized by RNN architectures, we make the following contributions.

(1) Focusing on *convolutional codes* we aim to decode them on the AWGN channel using RNN architectures. Efficient optimal decoding of convolutional codes has represented historically fundamental progress in the broad arena of algorithms; optimal bit error decoding is achieved by the 'Viterbi decoder' (Viterbi, 1967) which is simply dynamic programming or Dijkstra's algorithm on a specific graph (the 'trellis') induced by the convolutional code. Optimal block error decoding is the BCJR decoder (Bahl et al., 1974) which is part of a family of forward-backward algorithms. While early work had shown that vanilla-RNNs are capable in *principle* of emulating both Viterbi and BCJR decoders (Wang & Wicker, 1996; Sazl & Ik, 2007) we show empirically, through a careful construction of RNN architectures and training methodology, that neural network decoding is possible at very near optimal performances (both bit error rate (BER) and block error rate (BLER)). The key point is that we train a RNN decoder at a *specific* SNR and over *short information bit* lengths (100 bits) and show *strong generalization* capabilities by testing over a wide range of SNR and block lengths (up to 10,000 bits). The specific training SNR is closely related to the Shannon limit of the AWGN channel at the rate of the code and provides strong information theoretic collateral to our empirical results.

(2) *Turbo codes* are naturally built on top of convolutional codes, both in terms of encoding and decoding. A natural generalization of our RNN convolutional decoders allow us to decode turbo codes at BER comparable to, and at certain regimes, even *better* than state of the art turbo decoders on the AWGN channel. That data driven, SGD-learnt, RNN architectures can decode comparably is fairly remarkable since turbo codes already operate near the Shannon limit of reliable communication over the AWGN channel.

(3) We show the afore-described neural network decoders for both convolutional and turbo codes are *robust* to variations to the AWGN channel model. We consider a problem of contemporary

interest: communication over a "bursty" AWGN channel (where a small fraction of noise has much higher variance than usual) which models inter-cell interference in OFDM cellular systems (used in 4G and 5G cellular standards) or co-channel radar interference. We demonstrate empirically the neural network architectures can adapt to such variations and beat state of the art heuristics comfortably (despite evidence elsewhere that neural network are sensitive to models they are trained on (Szegedy et al., 2013)). Via an innovative local perturbation analysis (akin to (Ribeiro et al., 2016)), we demonstrate the neural network to have learnt sophisticated preprocessing heuristics in engineering of real world systems (Li et al., 2013).

## 2 RNN DECODERS FOR SEQUENTIAL CODES

Among diverse families of coding scheme available in the literature, *sequential coding* schemes are particularly attractive. They ($a$) are used extensively in mobile telephone standards including satellite communications, 3G, 4G, and LTE; ($b$) provably achieve performance close to the information theoretic limit; and ($c$) have a natural recurrent structure that is aligned with an established family of deep models, namely recurrent neural networks. We consider the basic sequential code known as *convolutional codes*, and provide a neural decoder that can be trained to achieve the optimal classification accuracy.

A standard example of a convolutional code is the *rate-1/2 Recursive Systematic Convolutional (RSC) code*. The encoder performs a forward pass on a recurrent network shown in Figure 1 on binary input sequence $\mathbf{b} = (b_1, \ldots, b_K)$, which we call *message bits*, with binary vector states $(s_1, \ldots, s_K)$ and binary vector outputs $(c_1, \ldots, c_K)$, which we call *transmitted bits* or a *codeword*. At time $k$ with binary input $b_k \in \{0, 1\}$ and the state of a two-dimensional binary vector $s_k = (s_{k1}, s_{k2})$, the output is a two-dimensional binary vector $c_k = (c_{k1}, c_{k2}) = (2b_k - 1, 2(b_k \oplus s_{k1}) - 1) \in \{-1, 1\}^2$, where $x \oplus y = |x - y|$. The state of the next cell is updated as $s_{k+1} = (b_k \oplus s_{k1} \oplus s_{k2}, s_{k1})$. Initial state is assumed to be 0, i.e., $s_1 = (0, 0)$.

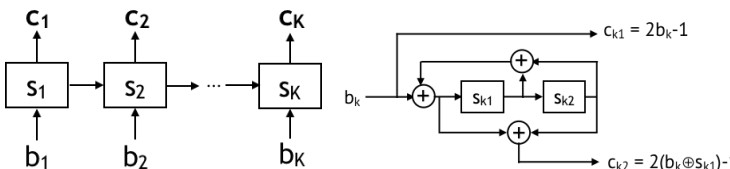

Figure 1: (Left) Sequential encoder is a recurrent network, (Right) One cell for a rate 1/2 RSC code

The $2K$ output bits are sent over a noisy channel, with the canonical one being the AWGN channel: the received binary vectors $\mathbf{y} = (y_1, \ldots, y_K)$, which are called the *received bits*, are $y_{ki} = c_{ki} + z_{ki}$ for all $k \in [K]$ and $i \in \{1, 2\}$, where $z_{ki}$'s are i.i.d. Gaussian with zero mean and variance $\sigma^2$. Decoding a received signal $y$ refers to (attempting to) finding the maximum a posteriori (MAP) estimate. Due to the simple recurrent structure, efficient iterative schemes are available for finding the MAP estimate for convolutional codes (vit; Bahl et al., 1974). There are two MAP decoders depending on the error criterion in evaluating the performance: bit error rate (BER) or block error rate (BLER).

BLER counts the fraction of blocks that are wrongly decoded (assuming many such length-$K$ blocks have been transmitted), and matching optimal MAP estimator is $\hat{\mathbf{b}} = \arg\max_{\mathbf{b}} \Pr(\mathbf{b}|\mathbf{y})$. Using dynamic programming, one can find the optimal MAP estimate in time linear in the block length $K$, which is called the *Viterbi algorithm*. BER counts the fraction of bits that are wrong, and matching optimal MAP estimator is $\hat{b}_k = \arg\max_{b_k} \Pr(b_k|\mathbf{y})$, for all $k = 1, \cdots, K$. Again using dynamic programming, the optimal estimate can be computed in $O(K)$ time, which is called the *BCJR algorithm*.

In both cases, the linear time optimal decoder crucially depends on the recurrent structure of the encoder. This structure can be represented as a hidden Markov chain (HMM), and both decoders are special cases of general efficient methods to solve inference problems on HMM using the principle of dynamic programming (e.g. belief propagation). These methods efficiently compute the exact posterior distributions in two passes through the network: the forward pass and the backward pass.

Our first aim is to train a (recurrent) neural network from samples, without explicitly specifying the underlying probabilistic model, and still recover the accuracy of the matching optimal decoders. At a high level, we want to prove by a constructive example that highly engineered dynamic programming can be matched by a neural network which only has access to the samples. The challenge lies in finding the right architecture and showing the right training examples.

**Neural decoder for convolutional codes**. We treat the decoding problem as a $K$-dimensional binary classification problem for each of the *message bits* $b_k$. The input to the decoder is a length-$2K$ sequence of received bits $\mathbf{y} \in \mathbb{R}^{2K}$ each associated with its length-$K$ sequence of "true classes" $\mathbf{b} \in \{0,1\}^K$. The goal is to train a model to find an accurate sequence-to-sequence classifier. The input $\mathbf{y}$ is a noisy version of the class $\mathbf{b}$ according to the rate-1/2 RSC code defined in earlier in this section. We generate $N$ training examples $(\mathbf{y}^{(i)}, \mathbf{b}^{(i)})$ for $i \in [N]$ according to this joint distribution to train our model.

We introduce a novel neural decoder for rate-1/2 RSC codes, we call N-RSC. It is two layers of bi-direction Gated Recurrent Units (bi-GRU) each followed by batch normalization units, and the output layer is a single fully connected sigmoid unit. Let $W$ denote all the parameters in the model whose dimensions are shown in Figure 2, and $f_W(\mathbf{y}) \in [0,1]^K$ denote the output sequence. The $k$-th output $f_W(\mathbf{y})_k$ estimates the posterior probability $\Pr(b_k = 1|\mathbf{y})$, and we train the weights $W$ to minimize the $L_w$ error with respect to a choice of a loss function $\ell(\cdot, \cdot)$ specified below:

$$\mathcal{L} = \sum_{i=1}^{N} \sum_{k=1}^{K} \ell(f_W(\mathbf{y}^{(i)})_k, b_k^{(i)}) . \tag{1}$$

As the encoder is a recurrent network, it is critical that we use recurrent neural networks as a building block. Among several options of designing RNNs, we make three specific choices that are crucial in achieving the target accuracy: $(i)$ bidirectional GRU as a building block instead of unidirectional GRU; $(ii)$ 2-layer architecture instead of a single layer; and $(iii)$ using batch normalization. As we show in Table 1 in Appendix C, unidirectional GRU fails because the underlying dynamic program requires bi-directional recursion of both forward pass and backward pass through the received sequence. A single layer bi-GRU fails to give the desired performance, and two layers is sufficient. We show how the accuracy depends on the number of layer in Table 1 in Appendix C. Batch normalization is also critical in achieving the target accuracy.

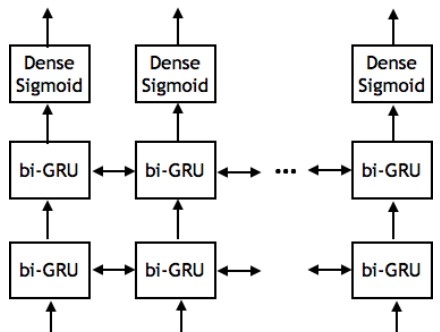

| Layer | Output dimension |
|---|---|
| Input | (K, 2) |
| bi-GRU | (K, 400) |
| Batch Normalization | (K, 400) |
| bi-GRU | (K, 400) |
| Batch Normalization | (K, 400) |
| Dense (sigmoid) | (K, 1) |

Figure 2: N-RSC: Neural decoder for RSC codes

**Training.** We propose two novel training techniques that improve accuracy of the trained model significantly. First, we propose a novel loss function guided by the efficient dynamic programming, that significantly reduces the number of training example we need to show. A natural $L_2$ loss (which gives better accuracy than cross-entropy in our problem) would be $\ell(f_W(\mathbf{y}^{(i)})_k, b_k^{(i)}) = (f_W(\mathbf{y}^{(i)})_k - b_k^{(i)})^2$. Recall that the neural network estimates the posterior $\Pr(b_k = 1|\mathbf{y}^{(i)})$, and the true label $b_k^{(i)}$ is a mere surrogate for the posterior, as typically the posterior distribution is simply not accessible. However, for decoding RSC codes, there exists efficient dynamic programming that can compute the posterior distribution exactly. This can significantly improve sample complexity of our training, as we are directly providing $\Pr(b_k = 1|\mathbf{y}^{(i)})$ as opposed to a sample from this distribution, which is $b_k^{(i)}$. We use a python implementation of BCJR in Taranalli (2015) to compute the posterior

distribution exactly, and minimize the loss

$$\ell(f_W(\mathbf{y}^{(i)})_k, b_k^{(i)}) = (f_W(\mathbf{y}^{(i)})_k - \Pr(b_k = 1|\mathbf{y}^{(i)}))^2 \; . \tag{2}$$

Next, we provide a guideline for choosing the training examples that improve the accuracy. As it is natural to sample the training data and test data from the same distribution, one might use the same noise level for testing and training. However, this is not reliable as shown in Figure 3.

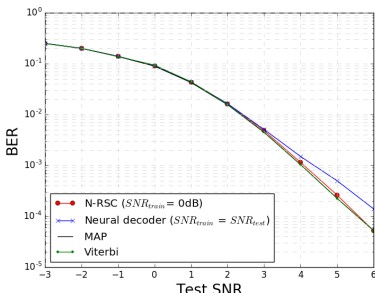

Figure 3: BER vs. test SNR for 0dB training and mismatched SNR training in decoding rate-1/2 RSC code of block length 100

Channel noise is measured by Signal-to-Noise Ratio (SNR) defined as $-10 \log_{10} \sigma^2$ where $\sigma^2$ is the variance of the Gaussian noise in the channel. For rate-1/2 RSC code, we propose using training data with noise level $\mathrm{SNR}_{\mathrm{train}} = \min\{\mathrm{SNR}_{\mathrm{test}}, 0\}$. Namely, we propose using training SNR matched to test SNR if test SNR is below $0dB$, and otherwise fix training SNR at $0dB$ independent of the test SNR. In Appendix D, we give a general formula for general rate-$r$ codes, and provide an information theoretic justification and empirical evidences showing that this is near optimal choice of training data.

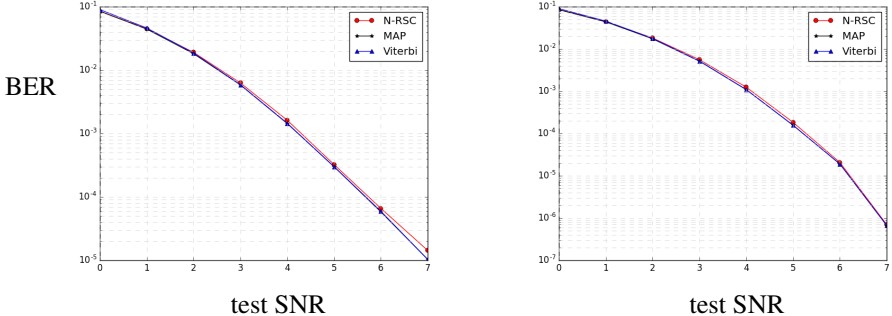

Figure 4: Rate-1/2 RSC code on AWGN. BER vs. SNR for block length (left) 100 and (right) 10,000

**Performance.** In Figure 4, for various test SNR, we train our N-RSC on randomly generated training data for rate-1/2 RSC code of block length 100 over AWGN channel with proposed training SNR of $\min\{\mathrm{SNR}_{\mathrm{test}}, 1\}$. We trained the decoder with Adam optimizer with learning rate 1e-3, batch size 200, and total number of examples is 12,000, and we use clip norm. On the left we show bit-error-rate when tested with length 100 RSC encoder, matching the training data. [1] We show that N-RSC is able to learn to decode and achieve the optimal performance of the optimal dynamic programming (MAP decoder) almost everywhere. Perhaps surprisingly, we show on the right figure that we can use the neural decoder trained on length 100 codes, and apply it directly to codes of length $10,000$ and still meet the optimal performance. Note that we only give $12,000$ training examples, while the number of unique codewords is $2^{10,000}$. This shows that the proposed neural decoder $(a)$ can generalize to unseen codeword; and $(b)$ seamlessly generalizes to significantly longer block lengths. More experimental results including other types of convolutional codes are provided in Appendix A.

---

[1]Source codes available in `https://github.com/yihanjiang/Sequential-RNN-Decoder`

We also note that training with $b_k^{(i)}$ in decoding convolutional codes also gives the same final BER performance as training with the posterior $\Pr(b_k = 1|\mathbf{y}^{(i)})$.

**Complexity.** When it comes to an implementation of a decoding algorithm, another important metric in evaluating the performance of a decoder is complexity. In this paper our comparison metrics focus on the BER performance; the main claim in this paper is that there is an alternative decoding methodology which has been hitherto unexplored and to point out that this methodology can yield excellent BER performance. Regarding the circuit complexity, we note that in computer vision, there have been many recent ideas to make large neural networks practically implementable in a cell phone. For example, the idea of distilling the knowledge in a large network to a smaller network and the idea of binarization of weights and data in order to do away with complex multiplication operations have made it possible to implement inference on much larger neural networks than the one in this paper in a smartphone (Hinton et al., 2015; Hubara et al., 2016). Such ideas can be utilized in our problem to reduce the complexity as well. A serious and careful circuit implementation complexity optimization and comparison is significantly complicated and is beyond the scope of a single paper. Having said this, a preliminary comparison is as follows. The complexity of all decoders (Viterbi, BCJR, neural decoder) is linear in the number of information bits (block length). The number of multiplications is quadratic in the dimension of hidden states of GRU (200) for the proposed neural decoder, and the number of encoder states (4) for Viterbi and BCJR algorithms.

**Turbo codes** are naturally built out of convolutional codes (both encoder and decoder) and represent some of the most successful codes for the AWGN channel (Berrou et al., 1993). A corresponding stacking of multiple layers of the convolutional neural decoders leads to a natural neural turbo decoder which we show to match (and in some regimes *even beat*) the performance of standard state of the art turbo decoders on the AWGN channel; these details are available in Appendix B. Unlike the convolutional codes, the state of the art (message-passing) decoders for turbo codes are not the corresponding MAP decoders, so there is no contradiction in that our neural decoder would beat the message-passing ones. The training and architectural choices are similar in spirit to those of the convolutional code and are explored in detail in Appendix B.

## 3 Non-Gaussian channels: Robustness and Adaptivity

In the previous sections, we demonstrated that the neural decoder can perform as well as the turbo decoder. In practice, there are a wide variety of channel models that are suited for differing applications. Therefore, we test our neural decoder under some canonical channel models to see how robust and adaptive they are. Robustness refers to the ability of a decoder trained for a particular channel model to work well on a differing channel model *without re-training*. Adaptivity refers to the ability of the learning algorithm to adapt and *retrain* for differing channel models. In this section, we demonstrate that the neural turbo decoder is both adaptive and robust by testing on a set of non-Gaussian channel models.

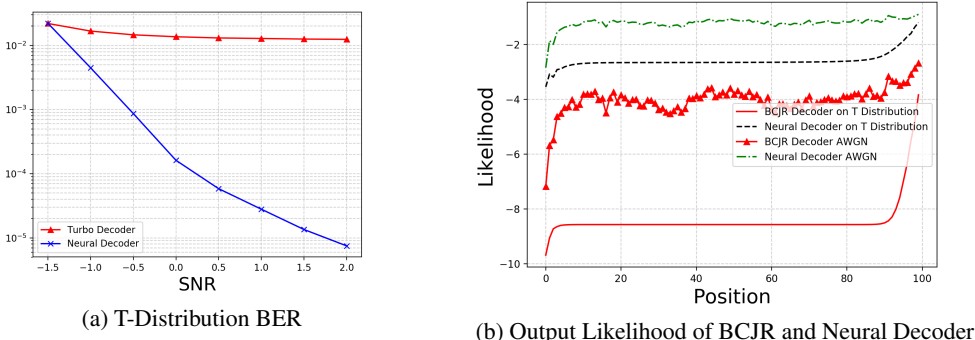

(a) T-Distribution BER

(b) Output Likelihood of BCJR and Neural Decoder

Figure 5: T-Distribution Performance

**Robustness.** The robustness test is interesting from two directions, other than obvious practical value. Firstly, it is known from information theory that Gaussian noise is the worst case noise

among all noise distributions with a given variance (Shannon, 1948; Lapidoth, 1996). Shannon showed in his original paper (Shannon, 1948) that among all memoryless noise sequences (with the same average energy), Gaussian noise is the worst in terms of capacity. After a long time, Lapidoth (1996) showed that for any finite block length, the BER achieved by the minimum distance decoder for any noise pdf is lower bounded by the BER for Gaussian noise under the assumption of Gaussian codebook. Since Viterbi decoding is the minimum distance decoder for convolutional codes, it is naturally robust in the precise sense above. On the other hand, turbo decoder does not inherit this property, making it vulnerable to adversarial attacks. We show that the neural decoder is more robust to a non-Gaussian noise, namely, t-distributed noise, than turbo decoder. Secondly, the robust test poses an interesting challenge for neural decoders since deep neural networks are known to misclassify when tested against small adversarial perturbations (Szegedy et al., 2013; Goodfellow et al., 2014). While we are not necessarily interested in adversarial perturbations to the input in this paper, it is important for the learning algorithm to be robust against differing noise distributions. We leave research on the robustness to small adversarial perturbations as a future work.

For the non-Gaussian channel, we choose the t-distribution family parameterized by parameter $\nu$. We test the performance of both the neural and turbo decoder in this experiment when $\nu = 3$ in Figure 5a and observe that the neural decoder performs significantly better than the standard Turbo decoder (also see Figure 16a in Appendix E). In order to understand the reason for such a bad performance of the standard Turbo decoder, we plot the average output log-likelihood ratio (LLR) $\log p(b_k = 1) - \log p(b_k = 1)$ as a function of the bit position in Figure 5b, when the input is all-zero codeword. The main issue for the standard decoder is that the LLRs are not calculated accurately (see Figure 16b in Appendix E): the LLR is exaggerated in the t-distribution while there is some exaggeration in the neural decoder as well, it is more modest in its prediction leading to more contained error propagation.

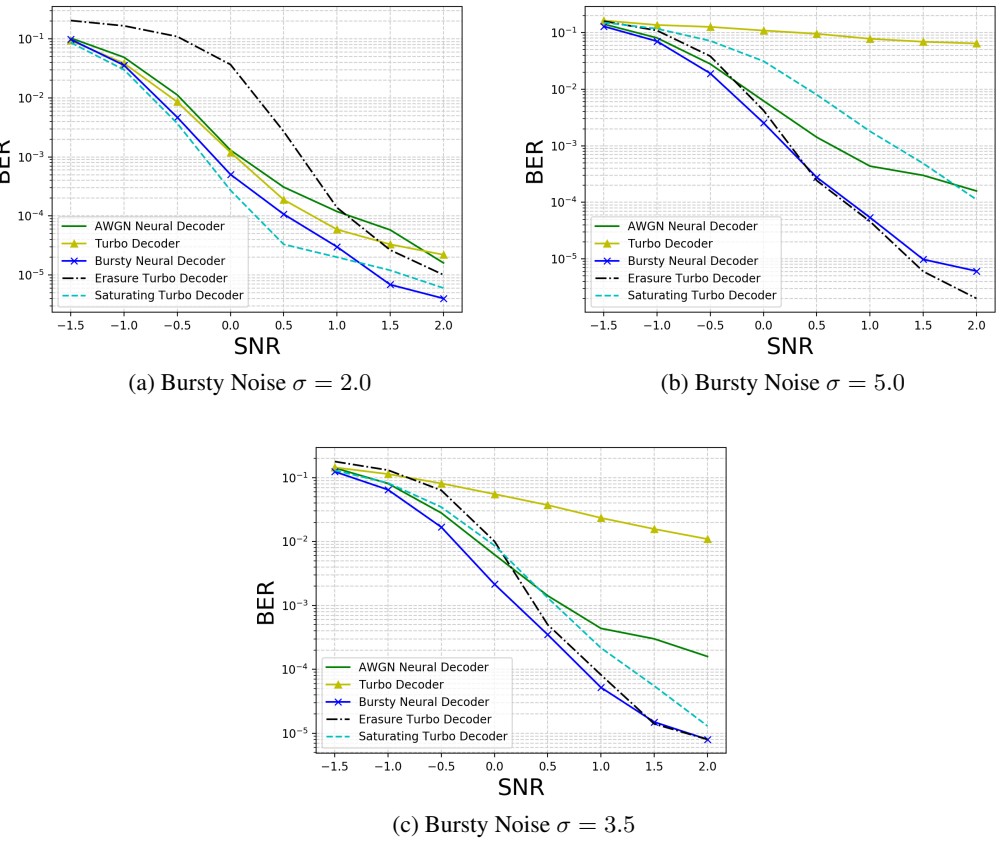

(a) Bursty Noise $\sigma = 2.0$

(b) Bursty Noise $\sigma = 5.0$

(c) Bursty Noise $\sigma = 3.5$

Figure 6: Neural Decoder Adaptivity under Different Bursty Noise Power

**Adaptivity.** A great advantage of neural channel decoder is that the neural network can learn a decoding algorithm even if the channel does not yield to a clean mathematical analysis. Consider a scenario where the transmitted signal is added with a Gaussian noise always, however, with a small probability, a further high variance noise is added. The channel model is mathematically described as follows, with $y_i$ describing the received symbol and $x_i$ denoting the transmitted symbol at time instant $i$: $y_i = x_i + z_i + w_i$, $z_i \sim N(0, \sigma^2)$, and $w_i \sim N(0, \sigma_b^2)$ with probability $\rho$ and $w_i = 0$ with probability $1 - \rho$, i.e., $z_i$ denotes the Gaussian noise whereas $w_i$ denotes the bursty noise.

This channel model accurately describes how radar signals (which are bursty) can create an interference for LTE in next generation wireless systems. This model has attracted attention in communications systems community due to its practical relevance (Sanders et al., 2013; Sanders, 2014). Under the aforesaid channel model, it turns out that standard Turbo coding decoder fails very badly (Safavi-Naeini et al., 2015). The reason that the Turbo decoder cannot be modified in a straight-forward way is that the location of the bursty noise is a latent variable that needs to be jointly decoded along with the message bits. In order to combat this particular noise model, we fine-tune our neural decoder on this noise model, initialized from the AWGN neural decoder, and term it the bursty neural decoder. There are two state-of-the-art heuristics (Safavi-Naeini et al., 2015): (a) erasure-thresholding: all LLR above a threshold are set to 0 (b) saturation-thresholding: all LLR above a threshold are set to the (signed) threshold.

We demonstrate the performance of our AWGN neural decoder (trained on Gaussian noise) as well as standard turbo decoder (for Gaussian noise) on this problem, shown in Figure 6 when $\sigma_b = 3.5, 2, 5$. We summarize the results of Figure 6: (1) standard turbo decoder not aware of bursty noise will result in complete failure of decoding. (2) standard neural decoder still outperforms standard Turbo Decoder. (3) Bursty-neural-decoder outperforms Turbo Decoder using both state-of-the-art heuristics at $\sigma_b^2 = 3.5$ and obtains performance approaching that of the better of the two schemes at other variances.

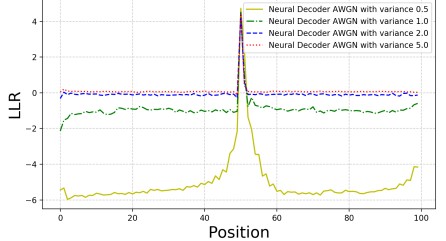

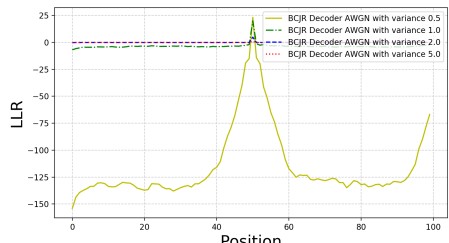

(a) Neural Decoder Positional likelihood under Bursty Noise

(b) BCJR Decoder Positional likelihood under Bursty Noise

Figure 7: Positional BER and Likelihood under Bursty Noise added in the 50th position

**Interpreting the Neural Decoder** We try to interpret the action of the neural decoder trained under bursty noise. To do so, we look at the following simplified model, where $y_i = x_i + z_i + w_i$ where $x_i, y_i, z_i$ are as before, but $w_i = B$ during the 50-th symbol in a 100-length codeword. We also fix the input codeword to be the all-zero codeword. We look at the average output LLR as a function of position for the one round of the neural decoder in Figure 7a and one round of BCJR algorithm in Figure 7b (the BER as a function of position is shown in Figure 17 in Appendix E). A negative LLR implies correct decoding at this level and a positive LLR implies incorrect decoding. It is evident that both RNN and BCJR algorithms make errors concentrated around the mid-point of the codeword. However, what is different between the two figures is that the scale of likelihoods of the two figures are quite different: the BCJR has a high sense of (misplaced) confidence, whereas the RNN is more modest in its assessment of its confidence. In the later stages of the decoding, the exaggerated sense of confidence of BCJR leads to an error propagation cascade eventually toggling other bits as well.

## 4 CONCLUSION

In this paper we have demonstrated that appropriately designed and trained RNN architectures can 'learn' the landmark algorithms of Viterbi and BCJR decoding based on the strong generalization capabilities we demonstrate. This is similar in spirit to recent works on 'program learning' in the literature (Reed & De Freitas, 2015; Cai et al., 2017). In those works, the learning is assisted significantly by a low level program trace on an input; here we learn the Viterbi and BCJR algorithms only by end-to-end training samples; we conjecture that this could be related to the strong "algebraic" nature of the Viterbi and BCJR algorithms. The representation capabilities and learnability of the RNN architectures in decoding existing codes suggest a possibility that new codes could be leant on the AWGN channel itself and improve the state of the art (constituted by turbo, LDPC and polar codes). Also interesting is a new look at classical multi-terminal communication problems, including the relay and interference channels. Both are active areas of present research.

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

APPENDIX

## A NEURAL DECODER FOR OTHER CONVOLUTIONAL CODES

The rate-1/2 RSC code introduced in Section 2 is one example of many convolutional codes. In this section, we show empirically that neural decoders can be trained to decode other types of convolutional codes as well as MAP decoder. We consider the following two convolutional codes.

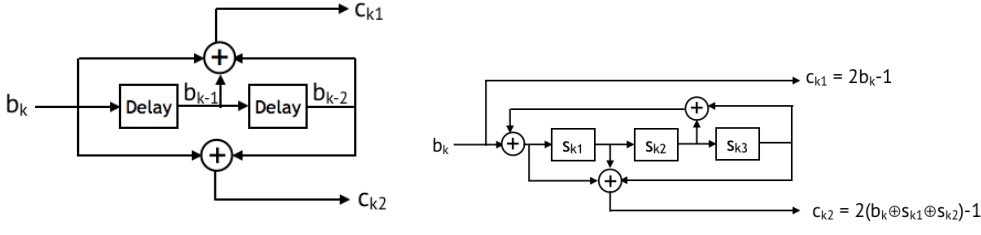

(a) Rate-1/2 non-recursive non-systematic convolutional code

(b) Rate-1/2 RSC code with state dimension 3

Figure 8: Examples of Rate 1/2 Convolutional code

Unlike the rate-1/2 RSC code in Section 2, the convolutional code in Figure 8($a$) is not recursive, i.e., state does not have a feedback. Also, it is non-systematic, i.e., the message bits can not be seen immediately from the coded bits. The convolutional code in Figure 8($b$) is another type of rate-1/2 RSC code with a larger state dimension (dimension 3 instead of 2).

Figure 8 show the architecture of neural network we used for the convolutional codes in Figure 8. For the code in Figure 8(a), we used the exact same architecture we used for the rate-1/2 RSC code in Section 2. For the code in Figure 8(b), we used a larger network (LSTM instead of GRU and 800 hidden units instead of 400). This is due to the increased state dimension in the encoder.

| Layer | Output dimension | | Layer | Output dimension |
|---|---|---|---|---|
| Input | (K, 2) | | Input | (K, 2) |
| bi-GRU | (K, 400) | | bi-LSTM | (K, 800) |
| Batch Normalization | (K, 400) | | Batch Normalization | (K, 800) |
| bi-GRU | (K, 400) | | bi-LSTM | (K, 800) |
| Batch Normalization | (K, 400) | | Batch Normalization | (K, 800) |
| Dense (sigmoid) | (K, 1) | | Dense (sigmoid) | (K, 1) |

Figure 9: Neural decoders for convolutional codes in (left) Figure 8 (a) and (right) Figure 8 (b)

For training of neural decoder in Figure 8(a), we used 12000 training examples of block length 100 with fixed SNR $0dB$. For training convolutional code (b), we used 48000 training examples of block length 500. We set batch size 200 and clip norm. The convolutional code (b) has a larger state space.

**Performance.** In Figures 10 , we show the BER and BLER of the trained neural decoder for convolutional code in Figure 8(a) under various SNRs and block lengths. As we can see from these figures, neural decoder trained on one SNR (0dB) and short block length (100) can be generalized to decoding as good as MAP decoder under various SNRs and block lengths. Similarly in Figure 11, we show the BER and BLER performances of trained neural decoder for convolutional code in Figure 8(b), which again shows the generalization capability of the trained neural decoder.

## B NEURAL DECODER FOR TURBO CODES

Turbo codes, also called parallel concatenated convolutional codes, are popular in practice as they significantly outperform RSC codes. We provide a neural decoder for turbo codes using multiple layers of neural decoder we introduced for RSC codes. An example of rate-1/3 turbo code is shown

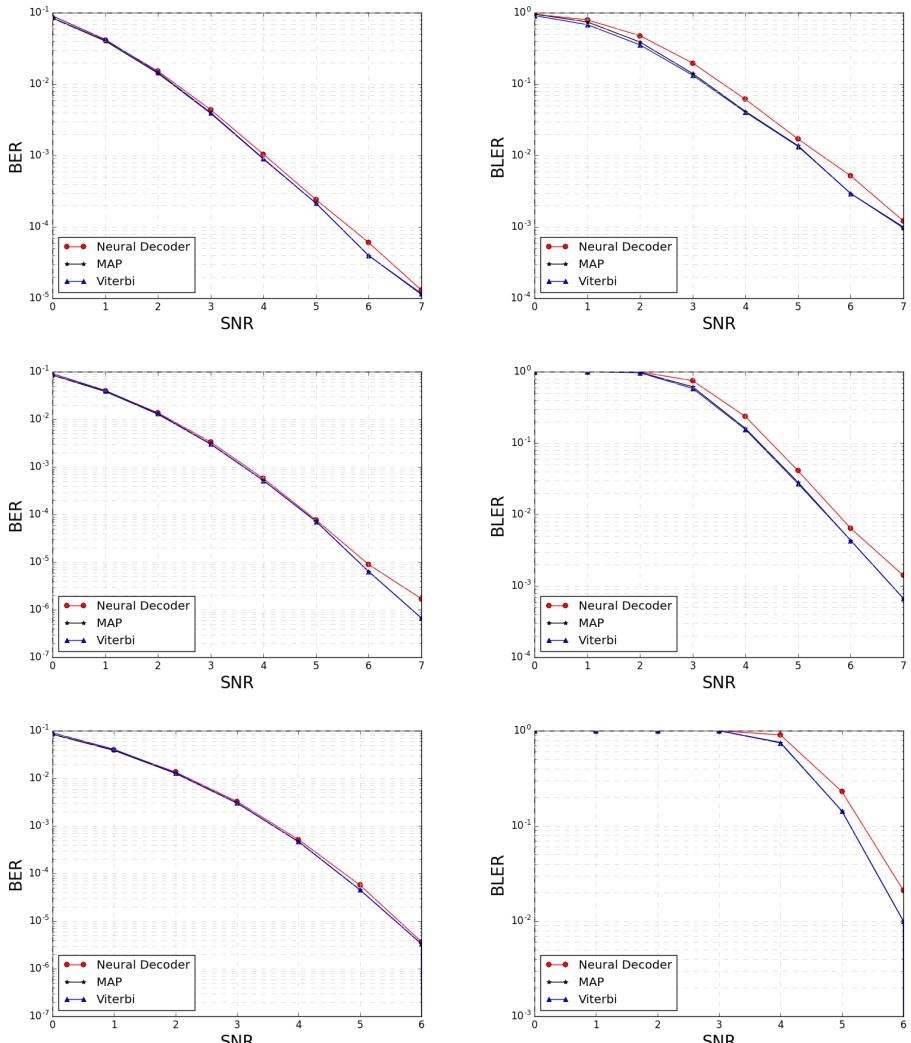

Figure 10: Rate-1/2 RSC code in Figure 8(a) on AWGN. (Left) BER and (Right) BLER vs. SNR for block length 100, 1000, and 10,000

in Figure 12. Two identical rate-1/2 RSC encoders are used, encoder 1 with original sequence $\mathbf{b}$ as input and encoder 2 with a randomly permuted version of $\mathbf{b}$ as input. Interleaver performs the random permutation. As the first output sequence $c_1(1)$ of encoder 1 is identical to the output sequence $c_1(2)$ of encoder 2, and hence redundant. So the sequence $c_1(2)$ is thrown away, and the rest of the sequences $(c_1(1), c_2(1), c_2(2))$ are transmitted; hence, rate is 1/3.

The sequences $(c_1(1), c_2(1), c_2(2))$ are transmitted over AWGN channel, and the noisy received sequences are $(y_1(1), y_2(1), y_2(2))$. Due to the interleaved structure of the encoder, MAP decoding is computationally intractable. Instead, an iterative decoder known as *turbo decoder* is used in practice, which uses the RSC MAP decoder (BCJR algorithm) as a building block. At first iteration the standard BCJR estimates the posterior $\Pr(b_k|y_1(1), y_2(1))$ with uniform prior on $b_k$ for all $k \in [K]$. Next, BCJR estimates $\Pr(b_k|\pi(y_1(1)), y_2(2))$ with the interleaved sequence $\pi(y_1(1))$, but now takes the output of the first layer as a prior on $b_k$'s. This process is repeated, refining the belief on what the codewords $b_k$'s are, until convergence and an estimation is made in the end for each bit.

**Training.** We propose a neural decoder for turbo codes that we call N-Turbo in Figure 12. Following the deep layered architecture of the turbo decoder, we stack layers of a variation of our N-RSC decoder, which we call N-BCJR. However, end-to-end training (using examples of the input sequence

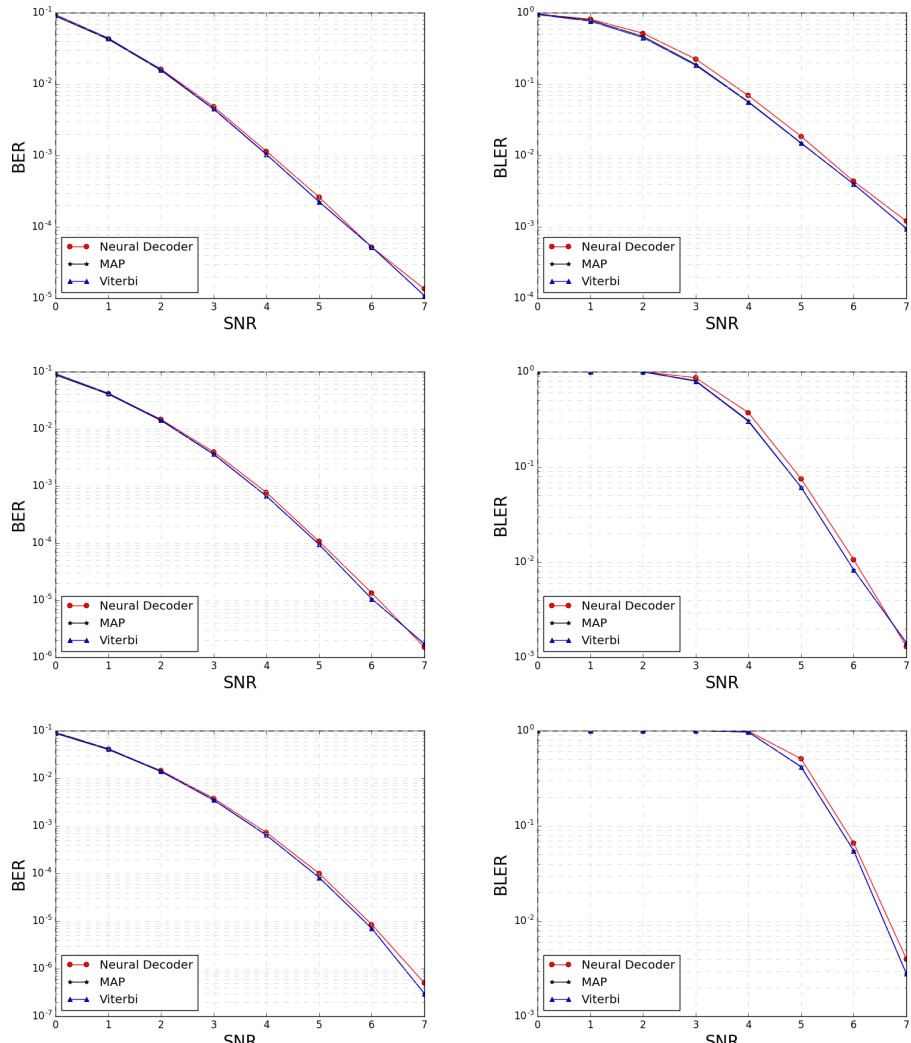

Figure 11: Rate-1/2 convolutional code in Figure 8(b) on AWGN. (Left) BER and (Right) BLER vs. SNR for block length 100, 1000, and 10,000

$\mathbf{y}^{(i)}$'s and the message sequence of $\mathbf{b}^{(i)}$'s of such a deep layers of recurrent architecture is challenging. We propose first training each layer separately, use these trained models as initializations, and train the deep layered neural decoder of N-Turbo starting from these initialized weights.

We first explain our N-BCJR architecture, which is a new type of N-RSC that can take flexible bit-wise prior distribution as input. Previous N-RSC we proposed is customized for uniform prior distribution. The architecture is similar to the one for N-RSC. The main difference is input size (3 instead of 2) and the type of RNN (LSTM instead of GRU). To generate $N$ training examples of $\{(noisy codeword, prior), posterior\}$, we generate $N/12$ examples of turbo codes. Then we ran turbo decoder for 12 component decoding - and collect input output pairs from the 12 intermediate steps of Turbo decoder, implemented in python Taranalli (2015) shown in Figure 13.

We train with codes with blocklength 100 at fixed SNR -1dB. We use mean squared error in (2) as a cost function.

To generate training examples with non-zero priors, i.e. example of a triplet (prior probabilities $\{\Pr(b_k)\}_{k=1}^{K}$, a received sequence $\mathbf{y}$, and posterior probabilities of the message bits $\{\Pr(b_k|\mathbf{y})\}_{k=1}^{K}$), we use intermediate layers of a turbo decoder. We run turbo decoder, and in each

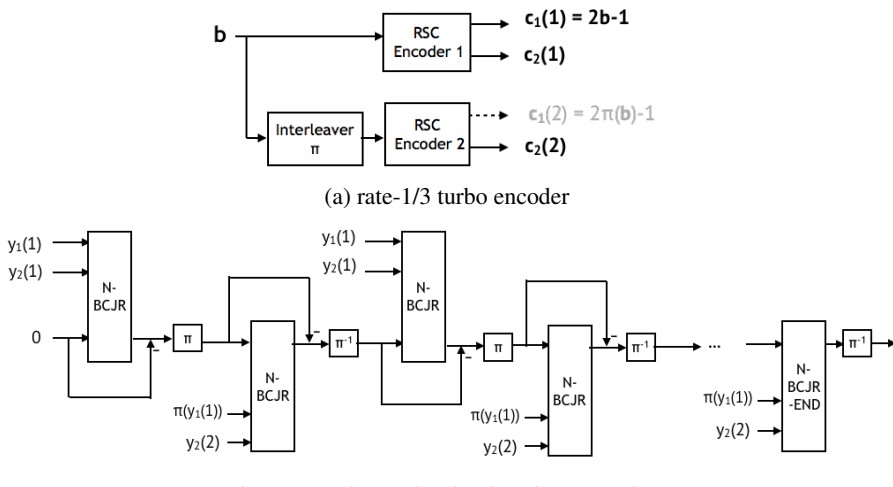

(a) rate-1/3 turbo encoder

(b) Proposed neural turbo decoder (N-Turbo)

Figure 12: rate-1/3 turbo encoder (top) and neural turbo decoder N-Turbo (bottom)

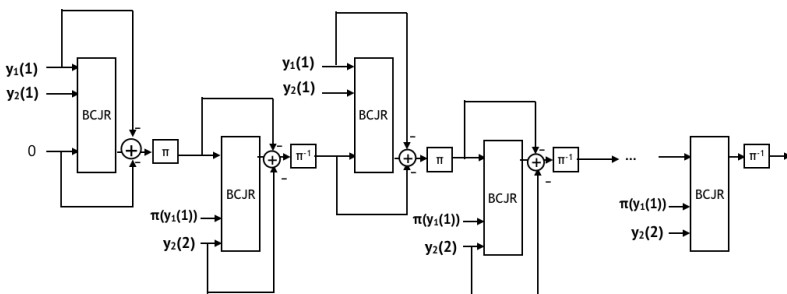

Figure 13: Turbo decoder

of the intermediate layers, we take as an example the triplet: the input prior probability, the input sequence, and the output of the BJCR layer. We fix training SNR to be -1dB. We stack 6 layers of BCJR decoder with interleavers in between. The last layer of our neural decoder is trained slightly differently to output the estimated message bit and not the posterior probability. Accordingly, we use binary crossentropy loss of as a cost function. We train each N-BCJR layer with 2,000 examples of length 100 turbo encoder, and in the end-to-end training of N-Turbo, we train with 1,000 examples of length 1,000 turbo encoder. We train with 10 epochs and ADAM optimizer with learning rate 0.001. For the end-to-end training, we again use a fixed SNR of noise (-1dB), and test on various SNRs. The choice of training SNR is discussed in detail in the Appendix D.

**Performance.** As can be seen in Figure 14, the proposed N-Turbo meets the performance of turbo decoder for block length 100, and in some cases, for test SNR= 2, it achieves a higher accuracy. Similar to N-RSC, N-Turbo generalizes to unseen codewords, as we only show $3,000$ examples in total. It also seamlessly generalizes in the test SNR, as training SNR is fixed at $-1$dB.

## C    OTHER NEURAL NETWORK ARCHITECTURES FOR N-RSC AND N-BCJR

In this section, we show the performances of neural networks of various recurrent network architectures in decoding rate-1/2 RSC code and in learning BCJR algorithm with non-zero priors. Table 1 shows the BER of various types of recurrent neural networks trained under the same condition as in N-RSC (120000 example, code length 100). We can see that BERs of the 1-layered RNN and single-directional RNN are order-wise worse than the one of 2-layered GRU (N-RSC), and two layers is sufficient. Table 2 shows the performance of neural networks of various recurrent network

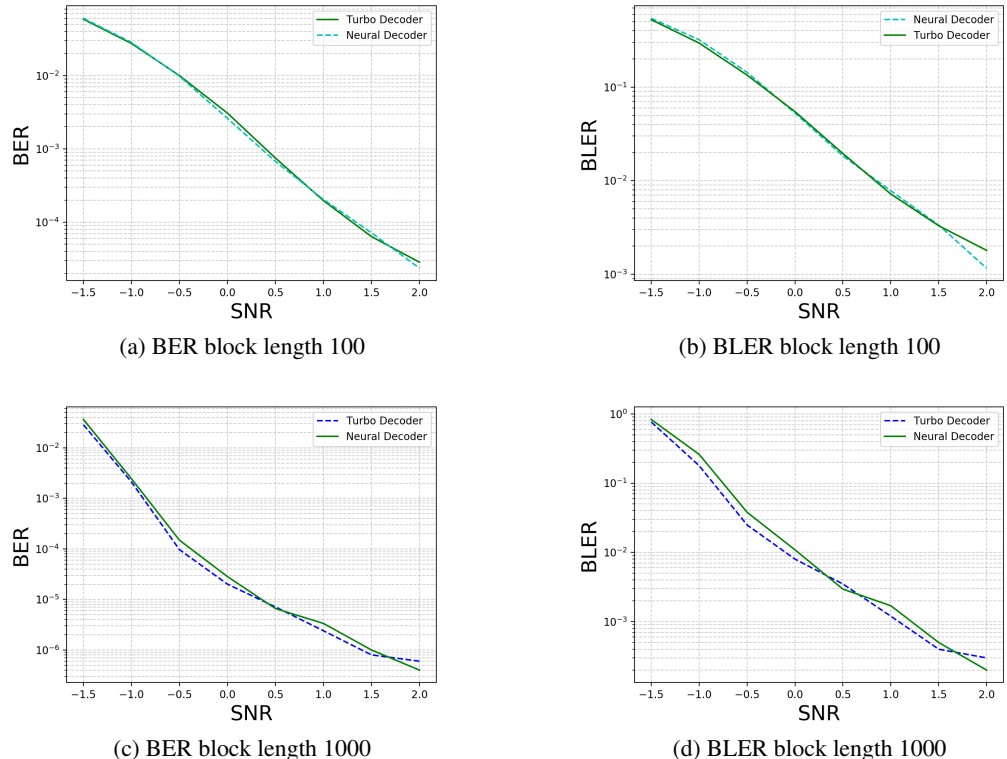

Figure 14: N-Turbo matches the performance of the Turbo decoder on AWGN.

architectures in BCJR training. Again, we can see that 2-layers are needed and single directional RNN does not work as well as bi-directional RNNs.

# D GUIDELINES FOR CHOOSING THE TRAINING SNR FOR NEURAL DECODERS

As it is natural to sample the training data and test data from the same distribution, one might use the same noise level for testing and training. However, this matched SNR is not reliable as shown in Figure 3. We give an analysis that predicts the appropriate choice of training SNR that might be different from testing SNR, and justify our choice via comparisons over various pairs of training and testing SNRs.

We conjecture that the optimal training SNR that gives best BER for a target testing SNR depends on the coding rate. A coding rate is defined as the ratio between the length of the message bit sequence $K$ and the length of the transmitted codeword sequence $\mathbf{c}$. The example we use in this paper is a rate $r = 1/2$ code with length of $\mathbf{c}$ equal to $2K$. For a rate $r$ code, we propose using training SNR according to

$$\text{SNR}_{\text{train}} = \min\{\text{SNR}_{\text{test}}, 10\log_{10}(2^{2r} - 1)\} , \qquad (3)$$

and call the knee of this curve $f(r) = 10\log_{10}(2^{2r} - 1)$ a threshold. In particular, this gives $\text{SNR}_{\text{train}} = \min\{\text{SNR}_{\text{test}}, 0\}$ for rate $1/2$ codes. In Figure 15 left, we train our neural decoder for RSC encoders of varying rates of $r \in \{1/2, 1/3, 1/4, 1/5, 1/6, 1/7\}$ whose corresponding $f(r) = \{0, -2.31, -3.82, -4.95, -5.85, -6.59\}$. $f(r)$ is plotted as a function of the rate $r$ in Figure 15 right panel. Compared to the grey shaded region of empirically observed region of training SNR that achieves the best performance, we see that it follows the theoretical prediction up to a small shift. The figure on the left shows empirically observed best SNR for training at each testing SNR for various rate $r$ codes. We can observe that it follows the trend of the theoretical prediction of a curve

| Depth | BER (at 4dB) | N (Training examples) | Hidden units |
|---|---|---|---|
| bi-LSTM-1 | 0.01376 | 12e+5 | 200 |
| bi-GRU-1 | 0.01400 | 12e+5 | 200 |
| uni-GRU-2 | 0.01787 | 12e+5 | 200 |
| bi-RNN-2 | 0.05814 | 12e+5 | 200 |
| bi-GRU-2 | 0.00128 | 12e+5 | 200 |
| bi-GRU-3 | 0.00127 | 12e+5 | 200 |
| bi-GRU-4 | 0.00128 | 12e+5 | 200 |
| bi-GRU-5 | 0.00132 | 12e+5 | 200 |

Table 1: BER (at 4dB) of trained neural decoders with different number/type of RNN layers on rate-1/2 RSC codes (blocklength 100) at SNR 4dB

| BCJR-like RNN Performance | | | |
|---|---|---|---|
| Model | Number of Hidden Unit | BCJR Val MSE | Turbo BER (Turbo 6 iters: 0.002) |
| BD-1-LSTM | 100 | 0.0031 | 0.1666 |
| BD-1-GRU | 100 | 0.0035 | 0.1847 |
| BD-1-RNN | 100 | 0.0027 | 0.1448 |
| BD-1-LSTM | 200 | 0.0031 | 0.1757 |
| BD-1-GRU | 200 | 0.0035 | 0.1693 |
| BD-1-RNN | 200 | 0.0024 | 0.1362 |
| SD-1-LSTM | 100 | 0.0033 | 0.1656 |
| SD-1-GRU | 100 | 0.0034 | 0.1827 |
| SD-1-RNN | 100 | 0.0033 | 0.2078 |
| SD-1-LSTM | 200 | 0.0032 | 0.137 |
| SD-1-GRU | 200 | 0.0033 | 0.1603 |
| SD-1-RNN | 200 | 0.0024 | 1462 |
| BD-2-LSTM | 100 | 4.4176e-04 | 0.1057 |
| BD-2-GRU | 100 | 1.9736e-04 | 0.0128 |
| BD-2-RNN | 100 | 7.5854e-04 | 0.0744 |
| BD-2-LSTM | 200 | 1.5917e-04 | 0.01307 |
| BD-2-GRU | 200 | 1.1532e-04 | 0.00609 |
| BD-2-RNN | 200 | 0.0010 | 0.11229 |
| SD-2-LSTM | 100 | 0.0023 | 0.1643 |
| SD-2-GRU | 100 | 0.0026 | 0.1732 |
| SD-2-RNN | 100 | 0.0023 | 0.1614 |
| SD-2-LSTM | 200 | 0.0023 | 0.1643 |
| SD-2-GRU | 200 | 0.0023 | 0.1582 |
| SD-2-RNN | 200 | 0.0023 | 0.1611 |

Table 2: MSE of trained neural models with different number/type of RNN layers in learning BCJR algorithm with non-zero priors

with a knee. Before the threshold, it closely aligns with the 45-degree line $\mathrm{SNR}_{\mathrm{train}} = \mathrm{SNR}_{\mathrm{test}}$. around the threshold, the curves become constant functions.

We derive the formula in (3) in two parts. When the test SNR is below the threshold, then we are targeting for bit error rate (and similarly the block error rate) of around $10^{-1} \sim 10^{-2}$. This implies that significant portion of the testing examples lie near the decision boundary of this problem. Hence, it makes sense to show matching training examples, as significant portion of the training examples will also be at the boundary, which is what we want in order to maximize the use of the samples. On the other hand, when we are above the threshold, our target bit-error-rate can be significantly smaller, say $10^{-6}$. In this case, most of the testing examples are *easy*, and only a very small proportion of the testing examples lie at the decision boundary. Hence, if we match training SNR, most of the examples will be wasted. Hence, we need to show those examples at the decision boundary, and we propose that the training examples from SNR $10 \log_{10}(2^{2r} - 1)$ should lie near the boundary. This is

a crude estimate, but effective, and can be computed using the capacity achieving random codes for AWGN channels and the distances between the codes words at capacity. Capacity is a fundamental limit on what rate can be used at a given test SNR to achieve small error. In other words, for a given test SNR over AWGN channel, Gaussian capacity gives how closely we can pack the codewords (the classes in our classification problem) so that they are as densely packed as possible. This gives us a sense of how decision boundaries (as measured by the test SNR) depend on the rate. It is given by the Gaussian capacity $\text{rate} = 1/2 \log(1 + SNR)$. Translating this into our setting, we set the desired threshold that we seek.

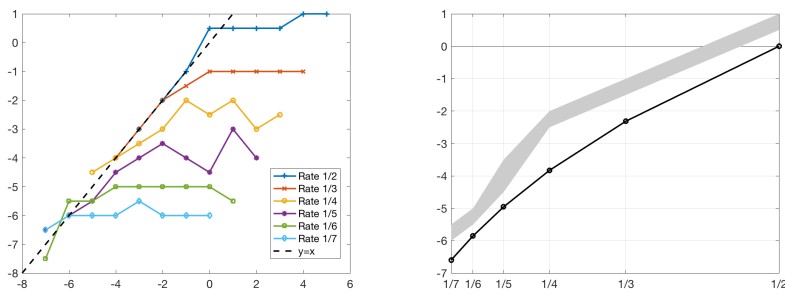

Figure 15: (Left) Best training SNR vs. Test SNR (Right) Best training SNR vs. code rate

# E SUPPLEMENTARY FIGURES FOR SECTION 3

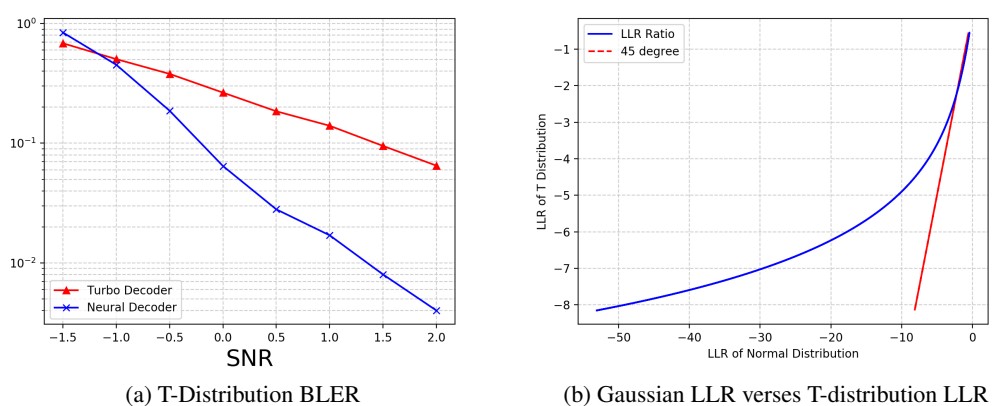

(a) T-Distribution BLER

(b) Gaussian LLR verses T-distribution LLR

Figure 16: T-Distribution Performance

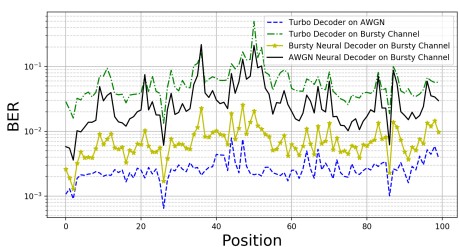

Figure 17: Turbo Decoder Positional BER log scale

