# OpenReview forum: "Communication Algorithms via Deep Learning"
_ICLR.cc/2018/Conference — Accept (Poster)_

### Official Review · AnonReviewer3 · 2017-11-27
**Deep learning for channel coding**

**Rating:** 2
**Confidence:** 4

**Review:**

In this paper the authors propose to use RNNs and LSTMs for channel coding. But I have the impression the authors completely miss the state of the art in channel coding and the results are completely useless for any current communication system. I believe that machine learning, in general, and deep learning, in particular, might be of useful for physical layer communications. I just do not see why it would be useful for channel coding over the AWGN channel. Let me explain.

If the decoder knows that the encoder is using a convolutional code, why does it need to learn the decoder instead of using the Viterbi or BCJR algorithms that are known to be optimal for sequences and symbols, respectively. I cannot imagine an scenario in which the decoder does not know the convolutional code that it is being used and the encoder sends 120,000 bits of training sequence (useless bits from information standpoint) for the decoder to learn it. More important question, do the authors envision that this learning is done every time there is a new connection or it is learnt once and for all. If it is learnt every time that would be ideal if we were discovering new channel codes everyday, clearly not the case. If we learnt it one and for all and then we incorporated in the standard that would only make sense if the GRU structure was computationally better than the BCJR or Viterbi. I would be surprise if it is. If instead of using 2 or 3 memories, we used 6-8 does 120,000 bits be good enough or we need to exponentially increase the training sequence? So the first result in the paper shows that a tailored structure for convolutional encoding can learn to decode it. Basically, the authors are solving a problem that does not need solving.

For the Turbocodes the same principle as before applies. In this case the comments of the authors really show that they do not know anything about coding. In Page 6, we can read: “Unlike the convolutional codes, the state of the art (message-passing) decoders for turbo codes are not the corresponding MAP decoders, so there is no contradiction in that our neural decoder would beat the message-passing ones”. This is so true, so I expected the DNN structure to be significantly better than turbodecoding. But actually, they do not. These results are in Figure 15 page 6 and the solution for the turbo decoders and the DNN architecture are equivalent. I am sure that the differences in the plots can be explained by the variability in the received sequence and not because the DNN is superior to the turbodecoder. Also in this case the training sequence is measured in the megabits for extremely simple components. If the convolutional encoders were larger 6-8 bits, we would be talking about significantly longer training sequences and more complicated NNs.

In the third set the NNs seems to be superior to the standard methods when burst-y noise is used, but the authors seems to indicate that that NN is trained with more information about these bursts that the other methods do not have. My impression is that the authors would be better of focusing on this example and explain it in a way that it is reproducible. This experiment is clearly not well explained and it is hard to know if there is any merit for the proposed NN structure.

Finally, the last result would be the more interesting one, because it would show that we can learn a better channel coding and decoding mechanism that the ones humans have been able to come up with. In this sense, if NNs can solve this problem that would be impressive and would turn around how channel coding is done nowadays. If this result were good enough, the authors should only focus in it and forget about the other 3 cases. The issue with this result is that it actually does not make sense. The main problem with the procedure is that the feedback proposal is unrealistic, this is easy to see in Figure 16 in which the neural encoder is proposed. It basically assumes that the received real-valued y_k can be sent (almost) noiselessly to the encoder with minimal delay and almost instantaneously. So the encoder knows the received error and is able to cancel it out. Even if this procedure could be implemented, which it cannot be. The code only uses 50 bits and it needed 10^7 iterations (500Mbs) to converge. The authors do not show how far they are from the Shannon limit, but I can imagine that with 50 bit code, it should be pretty far.

We know that with long enough LDPC codes we can (almost) reach the Shannon limit, so new structure are not needed. If we are focusing on shorter codes (e.g. latency?) then it will be good to understand why do we need to learn the channel codes. A comparison to the state of the art would be needed. Because clearly the used codes are not close to state of the art. For me the authors either do not know about coding or are assuming that we do not, which explains part of the tone of this review.

---

> ### Author Response · Authors · 2017-12-22
> **Response to AnonReviewr 3 (2/2)**
>
>
> Q5. NN is trained with more information about burst noise model than others? My impression is that the authors would be better of focusing on this example and explain it in a way that it is reproducible.
> A5. The two state-of-the-art heuristics - erasure thresholding and saturation thresholding (Safavi-Naeini et al. 2015) that we are comparing the NN decoder against to -  fully utilize the knowledge on the bursty channel model. Specifically, the threshold value in those methods is choosen based on the burst noise model. We believe the experiment is fully explained and entirely reproducible (we are also uploading our code base on Github after the ICLR review process is completed).  Perhaps the reviewer can be specific on exactly which parts of the experiment could use a better explanation.
>
> Q6. The feedback proposal is unrealistic. It basically assumes that the received real-valued y_k can be sent (almost) noiselessly to the encoder with minimal delay and almost instantaneously. So the encoder knows the received noise and is able to cancel it out.
> A6. (1) The AWGN channel with output feedback is the most classical of models in communication theory (studied by Shannon  himself in 1956. There has been a huge effort in the literature over the ensuing decades (the important  of which we have amply cited in our manuscript) and is of very basic importance to  multiple professional societies (including IEEE communication society and IEEE Information theory society). Although idealistic, it provides a valuable training ground to understand how to use feedback to more efficiently communicate.
> (2) The "W" in the  phrase AWGN (which is our channel model  refers to "white" which means the noise is memoryless across different time symbols. So even a single time step delay (not to mention "minimal delay") does not allow the "the encoder knows the received noise and is able to cancel it out."   Perhaps the reviewer would like to reconsider his/her ratiocination?

---

> ### Author Response · Authors · 2017-12-22
> **Repose to AnonReviewer 3 (1/2)**
>
> Below are detailed comments.
>
> Q1. Why should one use data to learn the Viterbi/BCJR/Turbo when we know them already?
> A1. This is because, we only know the optimal algorithms in simple settings and how to generalize them to more complicated or unknown channel models is sometimes unclear. We demonstrate that neural networks that learn from data can yield more robust and adaptable algorithms in those settings. This is the point of Section 3 and is elaborated below.
>
> Two advantages to RNN decoders, that go beyond mimicing Viterbi or BCJR:
> (1) Robustness: Viterbi and BCJR decoders are known to be vulnerable to changes in the channel, as those are highly tailored for the AWGN. We show in Section 3, via numerical experiments with T-dsitributed noise, that the neural network decoder trained on AWGN is much more robust against the changes in the channel. This makes, among other things, our neural network decoder much more attractive alternative to Viterbi or BCJR decoders in practice, where the channel model is not available.
> (2) Adaptivity: It is not easy to extend the idea of Viterbi decoder and iterative Turbo decoding beyond the simple convolutional codes and the standard Gaussian channel (or any other Discrete Memoryless Channel). On the other hand, our neural network decoder provides a new paradigm for decoding that can be applied to any encoder and any channel, as it learns from training examples. To showcase the power of this “adaptivity”, we show improved performance on bursty channels.
>
> A more stark example of the utility presents itself in the feedback channel. There exists no known practical encoding-decoding scheme for a feedback channel. Only because we have a neural network decoder that can adapt to any encoder, we are able to find a novel encoder (also NN based) that uses the feedback information correctly and achieves the performance significantly better than any other competing schemes. This would have not been possible without a NN decoder and the techniques we learned in training one to mimic the simple Viterbi.
>
> Q2. Learning is done every time there is a new connection or it is learnt once and for all?
> A2. We are not sure if we understand the question. The channel models in communication are *statistical* (and in particular the AWGN one) and codes are built to have a probabilistically good performance. The question of "learning done every time a connection is made" does not arise.
>
> Q3. If instead of using 2 or 3 memories, we used 6-8 does 120,000 bits be good enough or we need to exponentially increase the training sequence?
> A3. First note that even the Viterbi/BCJR decoder's computational complexity increases exponentially in the (memory) state dimension. While we have not tried a careful analysis on the complexity of neural decoders for codes with state dimension higher than 3, we observe the following: from dimension 2 to 3, we increased the size of neural decoder - from 2 layered bi-GRUs (200 hidden nodes) to 2 layered bi-LSTMs (400 hidden nodes).  The reason we havent explored a careful study of the memory size in neural network decoding is the following: modern coding theory recommends improving the 'quality' of the convolutional code not by increasing the memory state dimension, but via the 'turbo effect'.  The advantage of turbo codes over convolutional codes is that ​it uses convolutional codes with a short memory as the constituent codes, but the interleaver allows for very long range memory, that is naturally decoded via iterative methods. The end result is that turbo codes can be decoded with a far lesser decoding complexity than convolutional codes with a very long memory, for the same BER performance. Indeed, turbo codes have largely replaced convolutional codes in modern practice.
>
> Q4. In Figure 15 page 6, the solution for the turbo decoders and the DNN architecture are equivalent? I am sure that the differences in the plots can be explained by the variability in the received sequence and not because the DNN is superior to the turbodecoder.
> A4. Turbo codes have been used in every day cellular communication systems since 2000 and their decoders have been highly optimized.
> (1) A nontrivial improvement in BER performance over state of the art would be  considered a major development and discussion topic in standard body documents (we refer the reviewer to Annex B of 3GPP TS 36.101 (cited at the bottom of page 1 of our manuscript) from this summer for a feel for how large an impact on standard body decisions minor improvements to BER performance can have). Indeed, a "significantly better than turbo decoding" performance would qualify as a very major result in communication theory with correspondingly large impact on practice.
> (2) Our BER/BLER performance is averaged over 100000 blocks and the standard deviation is infinitesimally small. This is contradiction to the statement of being "sure that it can be can be explained by the variability in the received sequence."

---

> ### Comment · AnonReviewer2 · 2018-01-20
> **interesting or not?**
>
> As someone who has worked on coding theory for many years, I would like to add a comment, explaining
> why I found this paper very interesting and how it is related to this review.
>
> As mentioned, using density evolution we can design degree sequences for LDPCs that have thresholds that get very close to the Shannon limit. The only case where we can actually approach arbitrarily close is the erasure channel. For BIAWGN and BSC we can get quite close (but not actually arbitrarily close).
> However, for slightly more complicated channels we have no idea how to do that or even what the fundamental limits are (e.g. deletion channel).  I find this paper exciting because it defines a new family of possibilities in code and decoder design. It took us 50 years to go from Shannon's paper to modern LDPC and Turbo codes. So we should not expect that this paper beats LDPCs in their own game but rather as opening a new area of investigation.

---

> > ### Author Response · Authors · 2018-02-06
> > **Thank you to AnonReviewer2, and a few notes**
> >
> > We greatly thank AnonReviewer2 for his/her encouraging comments as well as proactively responding to AnonReviewer3. As coding/communication theorists ourselves, we can't agree more with AnonReviewer2: deep learning is a new and powerful tool for research in communications, e.g. to discover new communication algorithms. How to make deep learning to work as a tool for communications research is a vast new area of research.
> >
> > Our work is primarily on discovering new decoders for existing codes, and analyzing their robustness and adaptivity properties -- we agree with the reviewers that the result on the feedback channel is not as complete as our results in decoding sequential codes on the AWGN channel. We propose to remove this (small) section on the feedback channel in the final version of this paper and replace by a more detailed discussion on the adversarial training and testing performances (a major part of the review process).

---

### Official Review · AnonReviewer1 · 2017-11-28
**An interesting paper that brings in the tools of recursive neural networks to error-correcting codes for communication**

**Rating:** 6
**Confidence:** 4

**Review:**

Error-correcting codes constitute a well-researched area of study within communication engineering. In communication, messages that are to be transmitted are encoded into binary vector called codewords that contained some redundancy. The codewords are then transmitted over a channel that has some random noise. At the receiving end the noisy codewords are then decoded to recover the messages. Many well known families of codes exist, notably convolutional codes and Turbo codes, two code families that are central to this paper, that achieve the near optimal possible performance with efficient algorithms. For Turbo and convolutional codes the efficient MAP decodings are known as Viterbi decoder and the BCJR decoder. For drawing baselines, it is assumed that the random noise in channel is additive Gaussian (AWGN).

This paper makes two contributions. First, recurrent neural networks (RNN) are proposed to replace the Viterbi and BCJR algorithms for decoding of convolutional and Turbo decoders. These decoders are robust to changes in noise model and blocklength - and shows near optimal performance.

It is unclear to me what is the advantage of using RNNs instead of Viterbi or BCJR, both of which are optimal, iterative and runs in linear time. Moreover the authors point out that RNNs are shown to emulate BCJR and Viterbi decodings in prior works - in light of that, why their good performance surprising?

The second contribution of the paper constitutes the design and decoding of codes based on RNNs for a Gaussian channel with noisy feedback. For this channel the optimal codes are unknown. The authors propose an architecture to design codes for this channel. This is a nice step. However, in the performance plot (figure 8), the RNN based code-decoder does not seem to be outperforming the existing codes except for two points. For both in high and low SNR the performance is suboptimal to  Turbo codes and a code by Schalkwijk & Kailath. The section is also super-concise to follow. I think it was necessary to introduce an LSTM encoder - it was hard to understand the overall encoder. This is an issue with the paper - the authors previously mentioned (8,16) polar code without mentioning what the numbers mean.

However, I overall liked the idea of using neural nets to design codes for some non-standard channels. While at the decoding end it does not bring in anything new (modern coding theory already relies on iterative decoders, that are super fast), at the designing-end the Gaussian feedback channel part can be a new direction. This paper lacks theoretical aspect, as to no indication is given why RNN based design/decoders can be good. I am mostly satisfied with the experiments, barring Fig 8, which does not show the results that the authors claim.

---

> ### Author Response · Authors · 2017-12-22
> **Clarifications and the updated BER curve for our neural code on feedback channels (Figure 8)**
>
> Thank you for your comments.
>
> 1. Representability, Learnability and Generalization:
> There are three aspects to showing that a learning problem can be solved through a parametric architecture.
>
> (1) Representability: The ability to represent the needed function through a neural network. For Viterbi/BCJR algorithms, this representability was shown in prior work by handcrafting parameters that represent the Viterbi/BCJR algorithms. We note that neural networks with sufficient number of parameters can indeed represent any function through the universal approximation theorem for feedforward networks and RNNs (Cybenko,G.1989, Siegelmann,H.T.&Sontag,E.D.1995) and therefore this result is not that surprising.
>
> (2) Learnability: Can the required function be learnt directly through gradient descent on the observed data? For Viterbi and BCJR, learnability was neither known through prior work nor is it obvious. One of the main contributions of our work is that those algorithms can be learnt from observed data.
>
> (3) Generalization: Does the learnt function/algorithm generalize to unobserved data? We show this not only at the level of new unobserved codewords, but also show that the learnt algorithm trained on shorter blocks of length 100 can generalize well to longer blocks of length up to 10,000. Such generalization is rare in many realistic problems.
>
> To summarize, out of the three aspects, only representability was known from prior work (and, we agree with the reviewer, that it is the least surprising given universal representability). Learnability and generalization of the learnt Viterbi and BCJR algorithms to much larger block lengths are both unknown from prior art and they are surprising, and interesting in their own right. We note that Viterbi and BCJR algorithms are useful in machine learning beyond communications problem, representing dynamic programming and forward-backward algorithms, respectively.
>
> Peter Elias introduced convolutional codes in 1955 but efficient decoding through dynamic programming (Viterbi decoding) was only available in 1967 requiring mathematical innovation. We note that ability to learn the Viterbi algorithm from short block length data (which can be generated by full-search) and generalizing them to much longer blocks implies an alternative methodology to solve the convolutional code problem. Such an approach could have significant benefits in problems where corresponding mathematically optimal algorithms are not known at the moment.
>
> We demonstrate the power of this approach by studying the problem of channel-with-feedback where no good coding schemes are known despite 70 years of research.
>
> 2. Advantages of using RNNs instead of Viterbi or BCJR:
> There are two advantages to RNN decoders, that go beyond mimicing Viterbi/BCJR.
>
> (1) Robustness: Viterbi and BCJR decoders are known to be vulnerable to changes in the channel, as those are highly tailored for the AWGN. We show in Section 3, via numerical experiments with T-dsitributed noise, that the neural network decoder trained on AWGN is much more robust against the changes in the channel. This makes, among other things, our neural network decoder much more attractive alternative to Viterbi/BCJR decoders in practice, where the channel model is not available.
>
> (2) Adaptivity: It is not easy to extend the idea of Viterbi decoder and iterative Turbo decoding beyond the simple convolutional codes and the standard Gaussian channel (or any other Discrete Memoryless Channel). On the other hand, our neural network decoder provides a new paradigm for decoding that can be applied to any encoder and any channel, as it learns from training examples. To showcase the power of this “adaptivity”, we show improved performance on the bursty channel.
>
> A more stark example of the utility presents itself in the feedback channel. There exists no known practical encoding-decoding scheme for a feedback channel. Only because we have a neural network decoder that can adapt to any encoder, we are able to find a novel encoder (also neural network based) that uses the feedback information correctly and achieves the performance significantly better than any other competing schemes. This would have not been possible without a neural network decoder and the techniques we learned in training one to mimic the simple Viterbi.
>
> 3. Updated curve for new codes on AWGN channel with feedback:
> We have improved our encoder significantly by borrowing the idea of zero-padding from coding theory. In short, most of the errors occurs in the last bit, whose feedback information was not utilized by our encoder. We resolved this issue by padding a zero in the end of information bits (Hence, the codeword length is 3(K+1) for K information bits). This significantly improves the performance as shown in the new Figure 8. A full description of the encoder-decoder architecture is provided in Appendix D.
>
> 4. We replaced "(8,16) polar code" by “rate 1/2 polar code over 8 information bits”.

---

### Official Review · AnonReviewer2 · 2017-12-01
**RNNs decode convolutional codes very well**

**Rating:** 9
**Confidence:** 5

**Review:**

This paper shows how RNNs can be used to decode convolutional error correcting codes. While previous recent work has shown neural decoders for block codes results had limited success and for small block lengths.
This paper shows that RNNs are very suitable for convolutional codes and achieves state of the art performance for the first time.
The second contribution is on adaptivity outside the AWGN noise model. The authors show that their decoder performs well for different noise statistics outside what it was trained on. This is very interesting and encouraging. It was not very clear to me if the baseline decoders (Turbo/BCJR) are fairly compared here since better decoders may be used for the different statistics, or some adaptivity could be used in standard decoders in various natural ways.

The last part goes further in designing new error correcting schemes using RNN encoders and decoders for noisy feedback communication.
For this case capacity is known to be impossible to improve, but the bit error error can be improved for finite lenghts.
It seems quite remarkable that they beat Schalkwijk and Kailath and shows great promise for other communication problems.

The paper is very well written with good historical context and great empirical results. I think it opens a new area for information theory and communications with new tools.

My only concern is that perhaps the neural decoders can be attacked with adversarial noise (which would not be possible for good-old Viterbi ). It would be interesting to discuss this briefly.
A second (related) concern is the lack of theoretical understanding of these new decoders. It would be nice if we could prove something about them, but of course this will probably be challenging.

---

> ### Author Response · Authors · 2017-12-22
> **Adversarial noise attack and Fair comparison to baseline decoders**
>
> Thank you for your comments.
>
> 1. Neural decoders can be attacked with adversarial noise:
> This is a great point, which is related to the current ongoing advances in other areas of neural networks (e.g. classification). At a high level, there are two types of adversarial noise that can hurt our approach. The first one is poisoning training data. If we are training on data collected from real channels, an adversary who knows that we are training can intervene and add adversarial noise to make our trained decoder useless. This proposes an interesting game between the designer (us) and the attacker, in the form of how much noise power does the adversary need in order to make our decoder fail. This we believe is a fascinating research question, and we will add discussions in the final version of our manuscript.
> The second type of attack is adversarial examples, where at the test time an adversary changes the channel to make our decoder fail. In this scenario, both Viterbi and our decoder are vulnerable. Our numerical experiments on robustness is inspired by such scenarios, where we show that neural network decoders are more robust against such attacks (or natural dynamic changes) in Section 3.
>
> 2. Fair comparison to baseline decoders:
> There are two ways to run fair experiments on other channels, in our opinion. One is to mimic dynamic environment of real world by using encoder-decoders that are tailored for AWGN (both Turbo/BCJR and Neural Network decoder trained on AWGN) and see how robust it is against changes in the channel. This is the experiments we run with T-distribution. Another is, as suggested by the reviewer, to design decoders based on the new statistics of the channel that work well outside of AWGN. This is the experiments we run with bursty channels. We agree that these two experiments are addressing two different questions, but we believe we are fair in the comparisons to competing decoders within each setting.
>
> 3. Theoretical understanding of these neural decoders/coding schemes is a challenging but very interesting future research direction.

---

> > ### Public Comment · (anonymous) · 2018-01-04
> > **Viterbi decoder is vulnerable to adversarial examples?**
> >
> > Can you elaborate on how the Viterbi decoder is vulnerable to adversarial examples?

---

> > > ### Author Response · Authors · 2018-01-05
> > > **Re: Viterbi decoder is vulnerable to adversarial examples?**
> > >
> > > The answer is subtle and we explain in detail below.  We weren't as clear in our earlier response and we apologize for that.
> > >
> > > For an apples-to-apples comparison of two different (memoryless, random) noise sequences, let us keep the average energy (i.e., the expected value of the sum of squares of the noise values) to be the same. Then Shannon showed in his original 1948 paper that among all memoryless noise sequences (with the same average energy), Gaussian is the worst in terms of capacity. However it was not clear for a long time, if a decoder trained to be optimal for the Gaussian noise (i.e, the minimum distance decoder) would be robust to other noise pdfs. This was confirmed by a strong result of Lapidoth ’96 (Nearest Neighbor Decoding for Additive Non-Gaussian Noise Channels, IEEE Transactions on Information Theory): for any finite block length, the BER achieved by the minimum distance decoder for any noise pdf is *lower bounded* by the BER for Gaussian noise. Since Viterbi decoding is the minimum distance decoder for convolutional codes, it is naturally robust in the precise sense above.
> > > On the other hand, turbo decoder does not inherit this property, making it vulnerable to adversarial attacks. As can be seen in Section 3, the performance of turbo decoder (designed for the AWGN) under T distributed noise/bursty noise is extremely poor.
> > >
> > > When the noise is not Gaussian (which is the worst-case scenario), then there could be decoders that achieve much better performance. This is the sense in which our allusion to susceptibility of Viterbi decoding to adversarial/other noise pdfs was made. Specifically, we consider the practical scenario of bursty noise as an important example of non-Gaussian noise in this paper. Practical considerations suggest that we shouldn't keep the average noise comparable to Gaussian (which is the background noise and generally much smaller (20~50dB lower) than interference), so the apples-to-apples comparison setting discussed in the para is not as relevant. In this case, Viterbi decoder is not as effective as another decoder that harnesses the specific property of the bursty noise statistics -- indeed, as we see in Section 3, the neural network decoder is adaptive to this situation. Furthermore, when the noise is bursty, the turbo decoder with its constituent BCJR decoders is subject to severe error propagation that leads to a significant degradation in performance. We demonstrate that our learned neural network decoder outperforms well known hand-coded methods in the literature for this exact same setting.

---

> > > > ### Public Comment · (anonymous) · 2018-01-08
> > > > **Viterbi decoder is vulnerable to adversarial examples?**
> > > >
> > > > As far as I know, adversarial examples are inputs that we *expect* the model to process correctly, but it *surprisingly* doesn't. As an example, assume we have an image that is classified correctly. We *expect* small perturbations of that image to be correctly classified as well. A such perturbed image is called "adversarial example," if, to our *surprise*, it can fool the model.
> > > >
> > > > Now consider, say, Viterbi decoder and burst noise. Do we *expect* the model to process such cases correctly, and that it *surprisingly* doesn't?

---

> > > > > ### Author Response · Authors · 2018-01-10
> > > > > **Re: Viterbi decoder is vulnerable to adversarial examples?**
> > > > >
> > > > > Thanks for continuing this interesting discussion. Indeed, the presented definition of adversarial examples in the question is quite interesting: i.e., the departure from expectation. Measured that way, the example of Viterbi decoder operating on bursty signal will not qualify as an adversarial example (we had intended a different definition based on adversary injecting a signal to the channel whose goal is to make the decoder fail).
> > > > >
> > > > > However, if you take a Turbo decoder operating on a bursty signal, indeed one can justify that it is an adversarial example under the definition in question. This is because of the following: there is a theorem asserting that Gaussian noise is the worst case noise, i.e., once you fix the average power of the noise, the performance of a nearest-neighbor decoder will be worst if the noise is Gaussian. Therefore the performance on any other channel with the same average power should be *better.* However, in the case of the channel with Gaussian noise + bursty noise having the same average power, the performance of the *turbo* decoder is much worse than in the channel with only Gaussian noise. This is a case where the decoder behaves very differently from that of our expectation from the theoretical result; this is because the iterative BCJR decoder is not carrying out nearest neighbor decoding (unlike the Viterbi decoder).
> > > > >
> > > > > Furthermore, we would like to point out another nuance in adversarial examples. In the aforementioned cases, we are only changing the distribution of the channel (without looking at the data). A more stringent version, studied in the CS theory literature as worst-case noise, would allow the adversary to choose the noise after looking at the data input into the channel. Even in computer-vision, this is the usual definition, where the adversary looks at the image before injecting the noise. We would like to point out that for real communication problems, the coarse-grained adversary, which can only control the noise distribution maybe a bit more realistic.

---

> > > > > > ### Public Comment · (anonymous) · 2018-01-19
> > > > > > **Trubo decoder is vulnerable to adversarial examples?**
> > > > > >
> > > > > > Let me summarize what you said:
> > > > > > -- The theory says: for a decoder based on nearest neighboring (like Viterbi), Gaussian is the worst noise.
> > > > > > -- Performance of turbo decoder on Gaussian+burst noise is worse than Gaussian, because turbo is not a nearest neighbor decoder.
> > > > > >
> > > > > > Then you concluded: "This is a case where the decoder behaves very differently from that of our expectation from the theoretical result."
> > > > > >
> > > > > > Why did you *expect* Gaussian to be the worst noise for turbo, given that turbo is not nearest neighbor-based and hence the theory doesn't apply?
> > > > > >
> > > > > > I'm confused how does this suggest Gaussian+burst noise is an adversarial example for turbo decoder? We actually *don't expect* the decoder to work better for burst noise than Gaussian and the poor performance is *not surprising*.

---

> > > > > > > ### Author Response · Authors · 2018-01-22
> > > > > > > **Re: Turbo decoder is vulnerable to adversarial examples?**
> > > > > > >
> > > > > > > Thanks for your comments. Indeed, the Turbo decoder is not nearest neighbor, and therefore there is no theorem that the turbo decoder will perform better on every other noise distribution with the same variance. Indeed, if such was the case, there would be no way for the turbo decoder to do worse (since it is mathematically proven).
> > > > > > >
> > > > > > > On the other hand, Turbo codes do well on the AWGN channel, even when no theorem fully explains its performance. Thus this is an empirical fact, and sets up an empirical expectation that such good performance may extend to other distributions. Furthermore, Turbo decoder is deployed routinely in practice and its bad performance when the distribution is perturbed can affect co-existence with other technologies (like radar) and this has not been fully appreciated. And we show that it can be remedied.

---

> > > > > > > > ### Public Comment · (anonymous) · 2018-01-25
> > > > > > > > **Thanks!**
> > > > > > > >
> > > > > > > > Thanks!

---

### Public Comment · (anonymous) · 2017-11-02
**Complexity comparison**

Can you please elaborate on the complexity of the proposed neural decoder as compared to this of the Viterbi/BCJR decoder ?

Also, note a typo in Fig. 9: (a) and (b) figures should be exchanged.

---

> ### Author Response · Authors · 2017-11-06
> **Re: Complexity comparison**
>
>
> The complexity of all decoders (Viterbi, BCJR, Neural) is linear in the number of information bits (block length).
>
> The actual run times are hard to compare since ​some operations can be​ ​parallelized: e.g., matrix vector multiplications in the neural decoder can be easily parallelized in a GPU.
>
> Thanks for pointing out the typo.

---

> > ### Public Comment · (anonymous) · 2017-11-06
> > **Re: Complexity comparison**
> >
> > Thanks for your answer.
> >
> > Can you approximately calculate how many operations per information BIT you have in the neural decoer as compared to the viterbi decoder (like add-compare-select , multiplication, etc.) ?

---

> > > ### Author Response · Authors · 2017-11-09
> > > **Re: Complexity comparison**
> > >
> > > Thanks for your interest.
> > >
> > > The number of multiplications is quadratic in
> > > - the dimension of hidden states​ of GRU​ (200) for the proposed neural decoder, and
> > > - the number of encoder states (4) for Viterbi and BCJR.
> > >
> > > The number of add-compare-select units is
> > > - 0 for the proposed neural decoder, and
> > > - linear in the number of encoder states (4) for Viterbi.
> > >
> > > The dimension of hidden states in the GRU can be potentially reduced, using ideas such as network distillation. Apart from optimizing the size/complexity of the current neural decoder, significant parallelization is possible in the multiplicative units in the neural decoder, as well as pipelining. These designs in conjunction with a careful analysis of the fixed point arithmetic requirements of the different weights are under active research, and outside the scope of this paper.

---

### Public Comment · (anonymous) · 2017-11-06
**OTHER NEURAL NETWORK ARCHITECTURES FOR N-RSC AND N-BCJR**

For the results in Tables 1 and 2: are you using the code of Fig.1 or one of the codes of Fig.9 ?

Also, did you try similar analysis for codes with state dimension higher than 2 (or 3) ?
If not, what is your educated guess for the case of higher state dimensions?

---

> ### Author Response · Authors · 2017-11-17
> **Re: OTHER NEURAL NETWORK ARCHITECTURES FOR N-RSC AND N-BCJR**
>
> Tables 1 and 2 are for the code of Fig. 1.
>
> Re codes with state dimension higher than 2 (or 3):
>
> First note that the Viterbi/BCJR decoder's computational complexity increases exponentially in the (memory) state dimension. While we have not tried a careful analysis on the complexity of neural decoders for codes with state dimension higher than 3, we observe the following: from dimension 2 to 3, we increased the size of neural decoder - from 2 layered bi-GRUs (200 hidden nodes) to 2 layered bi-LSTMs (400 hidden nodes). It seems that the network has to scale as the state dimension increases, but we don't have a good guess of in what order it will scale.
>
> On a related note: ​Modern coding theory recommends improving the 'quality' of the convolutional code not by increasing the memory state dimension, but via the 'turbo effect'.  The advantage of turbo codes over convolutional codes is that ​it uses convolutional codes with a short memory as the constituent codes, but the interleaver allows for very long range memory, that is naturally decoded via iterative methods. The end result is that turbo codes can be decoded with a far lesser decoding complexity than convolutional codes with a very long memory, for the same BER performance. Indeed, turbo codes have largely replaced convolutional codes  in modern practice.

---

### Public Comment · (anonymous) · 2017-11-09
**"Leaning models" for "deterministic functions"?**

Interesting work; however, given that Viterbi algorithm, for example, is a simple (but elegant) well-defined algebraic function, isn't it expectable that a neural network with sufficient capacity would be able to approximate that?

Also, given the existing algorithm with efficient implementation, is it reasonable to replace it with an RNN? Will the time and memory complexity of such an RNN not be a major issue?

And, in general, is it a good practice to replace (or replicate) deterministic functions with "learning" data-driven models?

Of course, it would be very interesting if the ML model leads to the discovery of (or gives some insight into) a new class of coding schemes. But, this direction is not specifically pursued or examined in paper.

---

> ### Author Response · Authors · 2017-11-09
> **Re: "Leaning models" for "deterministic functions"?**
>
> Thanks for your interest.
>
> 1) Viterbi algorithm is not a simple deterministic function but an algorithm to find the shortest path on a directed graph (defined by the encoding structure as well as the number of information bits) with non-negative weights on edges (defined by the Gaussian noise samples).  In other words, it is the Dijkstra's shortest path algorithm (dynamic programming) on a specific graph that changes instance-by-instance (since the noise samples and number of information bits vary). While it is conceivable that Viterbi algorithm can be represented with a high capacity neural network, it is unclear if it can be learnt from data in reasonable training time. This is what we demonstrate.
>
> 2) Re memory/computation complexity: please see our response to "complexity comparison" below.
>
> 3) Why should one use data to learn the Viterbi algorithm (or BCJR or Turbo decoding) when we know them already? This is because, we only know the optimal algorithms in simple settings and how to generalize them to more complicated or unknown channel models is sometimes unclear. We demonstrate that neural networks that learn from data can yield more robust and adaptable algorithms in those settings; see Section 3.
>
> 4) We agree that it will be very interesting if the ML model leads to discovery of new class of coding schemes. Indeed, this is exactly what we show in Section 4, that for the Gaussian channel with feedback, the NN discovered codes beat the state-of-the-art codes.

---

### Public Comment · (anonymous) · 2018-01-05
**Fair comparison?**

The paper has not compared the complexity of decoders per information bit. This is a basic and essential comparison, especially in this case that efficient implementation is key for e.g., mobile devices. I think it is not fair to compare a highly more complex algorithm against a simpler one and claim superior performance.

---

> ### Author Response · Authors · 2018-01-05
> **Re: Fair comparison?**
>
> We accept this point in entirety. Both BER and complexity are important metrics of performance of a decoder. In this paper our comparison metrics have focused on the BER. We will make this point very clear in the revised paper. The main claim in the paper is that there is an alternative decoding methodology which has been hitherto unexplored and to point out that this methodology can yield excellent BER performance. Regarding the circuit complexity, we would like to point out that in computer vision, there have been many recent ideas to make large neural networks practically implementable in a cell phone. For example, the idea of distilling the knowledge in a large network to a smaller network and the idea of binarization of weights and data in order to do away with complex multiplication operations have made it possible to implement inference on much larger neural networks than the one in this paper in a smartphone. Such ideas can be utilized in our problem to reduce the complexity too. We would like to point out that a serious and careful circuit implementation complexity optimization and comparison is significantly complicated and submit that it is outside the scope of a single paper. Having said this, a preliminary comparison is discussed below with another anonymous reviewer, but we provide it here for completeness:
>
> The number of multiplications is quadratic in
> - the dimension of hidden states​ of GRU​ (200) for the proposed neural decoder, and
> - the number of encoder states (4) for Viterbi and BCJR.
>
> The number of add-compare-select units is
> - 0 for the proposed neural decoder, and
> - linear in the number of encoder states (4) for Viterbi.
>
> Apart from optimizing the size/complexity of the current neural decoder, significant parallelization is possible in the multiplicative units in the neural decoder, as well as pipelining. These designs in conjunction with a careful analysis of the fixed point arithmetic requirements of the different weights are under active research, and outside the scope of this paper.
>
> More generally circuit implementation complexity improves with time due to both hardware design and component improvements. This has been the case throughout the history of reliable communication: promising algorithms are first proposed and then followed by a surge of research in efficient circuit implementation. A case in point is the recently proposed polar codes where significant recent research has made its decoding competitive in practice (and indeed has been accepted for parts of the 5G wireless standard earlier in the summer of 2017).

---

> > ### Public Comment · (anonymous) · 2018-01-07
> > **Fair comparison**
> >
> > Ideas, such as distillation or binarization, have been mostly applied to image data and is not immediately clear whether they will work on other data types as well. Honestly, I don't think they will reduce complexity without significantly losing performance, unless experimental results will prove otherwise.
> >
> > Also, compared to vision, there is a really real-time constraint in decoding, as typically millions of bits needs to be decoded every second...
> >
> > I do like the idea of the paper (this is why I'm commenting here after all), but not sure if it will be practical.

---

> > > ### Author Response · Authors · 2018-01-10
> > > **Re: Fair comparison**
> > >
> > > Thanks for your interest. We agree that latency is an important factor and the throughput demand in decoding is even more aggressive than in computer vision. We would like to point out that our neural networks are much more shallow (2 layers) compared to the 1000 layers employed in vision.   It is not immediately obvious how the computational tradeoffs will play out in this case. This is beyond the scope of the present paper and an important direction for further research.

---

### Public Comment · (anonymous) · 2018-01-08
**A note about generalization**

Neat idea and results, and I'm excited to see more people working on this.

You wrote "generalization is difficult [when using neural networks to decode non-sequential codes]", and the focus of your paper is on convolutional codes.

However, there actually have been a few papers recently that get it to work for arbitrary linear codes and longer polar codes:

"Learning to Decode Linear Codes Using Deep Learning"-  https://arxiv.org/abs/1607.04793
"Neural Offset Min-Sum Decoding" - https://arxiv.org/abs/1701.05931
"Deep Learning Methods for Improved Decoding of Linear Codes" - https://arxiv.org/abs/1706.07043
"Scaling Deep Learning-based Decoding of Polar Codes via Partitioning" - https://arxiv.org/abs/1702.06901
"improved Polar Decoder Using Deep Learning" - https://www.researchgate.net/publication/321122117_Improved_polar_decoder_based_on_deep_learning

---

> ### Author Response · Authors · 2018-01-10
> **Re: A note about generalization**
>
> Thank you for the references. We will cite them appropriately in the final version.

---

### Decision · Program_Chairs · 2018-01-29
**ICLR 2018 Conference Acceptance Decision**

**Decision:**

Accept (Poster)

**Comment:**

This paper studies trainable deep encoders/decoders in the context of coding theory, based on recurrent neural networks. It presents highly promising results showing that one may be able to use learnt encoders and decoders on channels where no predefined codes are known.

Besides these encouraging aspects, there are important concerns that the authors are encouraged to address; in particular, reviewers noted that the main contribution of this paper is mostly on the learnt encoding/decoding scheme rather than in the replacement of Viterbi/BCJR. Also, complexity should be taken into account when comparing different decoding schemes.

Overall, the AC leans towards acceptance, since this paper may trigger further research in this direction.